# communications
# engineering

# A Quaternary mixed oxide protective scaffold for ruthenium during oxygen evolution reaction in acidic media

Alexis Piñeiro-García[1], Xiuyu Wu[1], Mouna Rafei[1], Paul Jonathan Mörk[1,2] & Eduardo Gracia-Espino [1✉]

Proton exchange membrane water electrolysis is widely used in hydrogen production, but its application is limited by significant electrocatalyst dissolution at the anode during the oxygen evolution reaction (OER). The best performing electrocatalysts to date are based on ruthenium and iridium oxides, but these experience degradation even at moderate cell potentials. Here we investigate a quaternary Sn-Sb-Mo-W mixed oxide as a protective scaffold for ruthenium oxide. The acid-stable mixed oxide consists of an interconnected network of nanostructured oxides capable of stabilizing ruthenium into the matrix (Ru-MO). In combination with titanium fibre felt, we observed a lower degradation in the oxygen evolution reaction activity compared to unprotected ruthenium oxide after the electrochemical stress test. The superior stability of Ru-MO@Ti is attributed to the presence of MO which hinders the formation of reactive higher valence ruthenium ($Ru^{+8}$). Our work demonstrates the potential of multi-metal oxides to extend the lifetime of the OER active metal and the titanium support.

[1] Department of Physics, Umeå University, SE-901 87 Umeå, Sweden. [2] Faculty of physics and astronomy, Julius-Maximilians-Universität Würzburg, Würzburg, Germany. ✉email: eduardo.gracia@umu.se

The production of hydrogen gas using solely renewable electricity has remained as a desirable solution to the harmful use of fossil fuels since the 70's, when the term hydrogen economy was coined[1]. Hydrogen production via proton exchange membrane (PEM) water electrolysis is seen as a promising solution due to its ultra-pure hydrogen generation, high current densities, high efficiency, fast response, and small footprint[2–4]. However, its biggest limitation is the substantial electrocatalyst dissolution seen at the anode during the oxygen evolution reaction (OER)[5]. Currently, efforts to extend the lifetime of noble metal catalyst are in the spotlight and some advances have been obtained in this field[6–16], but state-of-the-art OER electrocatalysts are still based on iridium and ruthenium, either in the pure oxide form or as engineered composites. However, these electrocatalysts still dissolve at moderate current densities resulting in poor operational stability. Therefore, a compromise between stability and activity must be made. Naturally, there are several approaches to extend the lifetime of such electrocatalysts where the most attractive approaches are the production of $RuO_2/IrO_2$ mixed-oxide phase[6], $RuO_2$ mixed with earth abundant metal oxides[7], polychlores oxides[8], and perovskites-based oxides[9,12]. Among these, earth abundant mixed oxides that stabilize Ru stand out as a promising option to overcome Ru dissolution during OER conditions. A variety of metal oxides have been investigated as supports and stabilizing frameworks for $RuO_2$, such as $CrO_x$[5], $Sb-SnO_2$[6], $TaO_x$[10], $TiO_2$[17], $WO_3$[18], and pyrochlore complex[9]. This strategy brings benefits such as improved stability in acidic media, large-scale production, low cost of electrodes, and in some cases improved OER activity.

Here, we developed a multicomponent mixed-oxide matrix that can host OER active metals and protect them against early dissolution under harsh acidic conditions and high anodic potentials. The quaternary mixed oxide comprises a mixture of Sn, Sb, Mo, and W interconnected oxides forming a continues porous structure. Solution precursor plasma spraying (SPPS) was employed as a simple and versatile method for the scalable production of multicomponent oxides. The mixed oxide (MO) was deposited on fluorine doped tin oxide (FTO) coated glass where it was probed to be stable in acidic conditions (from pH = 2 to 0) under applied anodic potentials (1.8 − 2 V vs RHE). Then, the MO was employed as protective scaffold where $RuO_2$ was anchored (Ru-MO, 2.64% w/w Ru). We identified that Ru dissolution proceeded through a second-order reaction, and the kinetics of the deactivation was used to guide the incorporation of Ru. Finally, titanium fiber felt (a typical gas diffusion layer) was used as substrate (Ru-MO@Ti) where only a decrease of 20% in the OER activity was observed after 2000 CVs (1.2–1.8 V vs RHE). We attributed the superior stability of the new material to the presence of the MO which hinders the formation of $Ru^{+8}$, and thus reducing the corrosion rate. The quaternary mixed-oxide matrix proposed in this work can help to extend the lifetime of both $RuO_2$ and titanium fiber felt used in water electrolysis PEM cells.

## Results and discussion

Sn-Sb-Mo-W mixed-oxide (MO) coatings were produced on FTO glass via solution precursor plasma spraying. The coating comprises a mixture of $SnO_x$, $SbO_x$, $WO_x$, and $MoO_x$. The molar ratio of the metal precursors Sn:Sb:Mo:W was set to 5:5:18:18. This particular ratio was selected based on a previous theoretical report[17,19,20], and our own preliminary experimental studies (Fig. S1), both indicating that binary metallic oxides of $WO_x$ and $MoO_x$ (W:Mo = 1:1 molar ratio), $SnO_2$ and $Sb_2O_3$ (Sn:Sb = 1:1 molar ratio), as well as $WO_x$ and $SnO_2$ (W:Sn = 18:1 molar ratio) can form stable mixed oxides under acidic and anodic potentials. Figure 1a shows a schematic of the coating process where the

solution precursor is introduced into the plasma plume via an external nozzle. The generated droplets travel inside the plasma plume where they undergo solvent evaporation, precursor decomposition, and particle formation during their trajectory before reaching the substrate[21,22]. SPPS process typically forms spherical particles, hollowed or broken-shell particles, producing coatings that are generally porous. The coating is then formed by successive deposition of layers until a desirable thickness is achieved. Afterwards, the as-produced coatings are subjected to an oxidation process in air at 500 °C for 2 h, transforming the initial amorphous coating into a mixture of metal oxides with distinct oxidation states (Fig. 1b).

The morphology and chemical composition of the pristine MO coatings were characterized before the electrochemical analysis. Scanning electron microscopy (SEM) images of the MO coatings, Fig. 2a, reveal a homogeneous but semi-continuous coating constituted by fissures and cavities. Such defects are likely formed due to solvent evaporation and reactant decomposition (e.g., metal hydroxides trapped during spaying) during the spraying or annealing processes. Additionally, some micrometer spherical particles and rod-like structures (Fig. 2a, indicated by the arrows) were observed through the surface of the coating. Energy dispersive X-ray (EDX) elemental mapping (Fig. 2b) revealed a homogenous distribution of transition metals through the entire surface, irrespective of the observed morphology (surface coating, spheres, rods, voids, etc.). This information points towards a favorable interaction among the oxides, either due to an atomically interconnected network of oxides (forming binary, ternary, or quaternary oxides) or a mixture of single-metal oxide nanocrystallites creating heterojunctions. XRD patterns point towards a mixture of single-metal oxides (see Fig. S2a), however, at the moment we are unable to disregard the formation of multimetallic oxides, as well as doping withing the oxides due to the complex characteristics of the coating.

The chemical state of the elements was investigated by X-ray photoelectron spectroscopy (XPS). The high resolution XPS corelevel spectra of W 4 f and Mo 3d (Fig. 2c, right to left) show clear characteristics peaks corresponding to metal oxides with oxidation state of +4 ($WO_2$, $MoO_2$) and +6 ($WO_3$, $MoO_3$)[18,23]. On the other hand, the Sn 3d region exhibited a strong contribution from Na KLL Auger electron (region highlighted in green) from the tungsten precursor ($Na_2WO_4$)[24]. The peak fitting was completed by keeping a 3:2 area ratio between the Sn $3d_{5/2}$ and the Sn $3d_{3/2}$ features, as well as a spin-orbital split distance of 8.4 eV, as previously reported[24,25]. With this, the Sn 3d region showed features associated to metallic tin ($Sn^{+0}$) and $SnO_2$ ($Sn^{+4}$). Finally, the analysis of the region corresponding to Sb 3d and O 1 s was performed. In this case due to the overlapping of the binding energies, the peak fitting was carried out by first using the Sb $3d_{3/2}$ peak as reference with a spin-orbital split distance of 9.4 eV and a 3:2 area ratio with respect to Sb $3d_{5/2}$[26]. Then, oxygen-related peaks were added until completing the experimental data of the O 1 s region. The peaks found in the Sb 3d region correlate with $Sb_2O_3$[26], while the O 1 s deconvolution indicates peaks associated to Sn-O bonds[27], lattice oxygen species ($O_{lattice}$)[28], and adsorbed-water molecules ($O_{H2Oads}$)[29]. No peak associated to vacancies was found in the O 1 s spectra. Based on the XPS results, the MO coating consists of individual metals and metal oxides with multiple oxidation states. Interestingly, a highly homogeneous metal distribution with no sign of segregations was observed by EDX mapping (Fig. 2b), therefore we expect to observe synergy effect due to doping and well-formed heterojunctions.

**Stability in acidic media and anodic potentials.** The stability of the MO coatings at different pH (MO@pH2, MO@pH1 and MO@pH0) was first evaluated by cycling voltammetry (20 CV

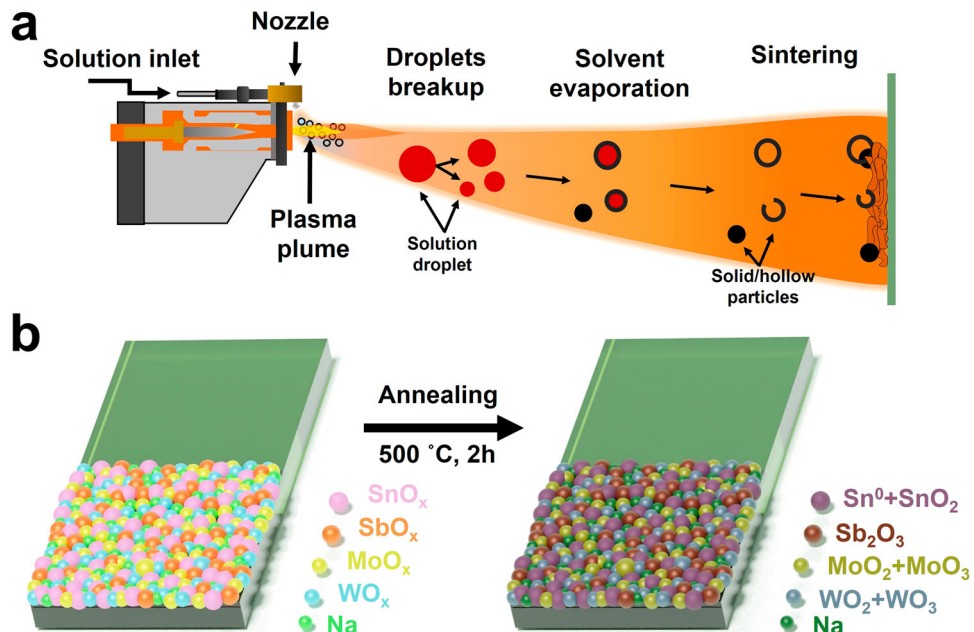

**Fig. 1 Production of the quaternary mixed oxide. a** Schematic of the plasma torch and coating formation. The solution precursor is fed into the plasma plume via an external nozzle. A raster spraying pattern is used to form the coating using a robotic arm. **b** The as-sprayed amorphous coatings are subjected to an oxidation procedure in air to form a coating constituted by various metal oxides.

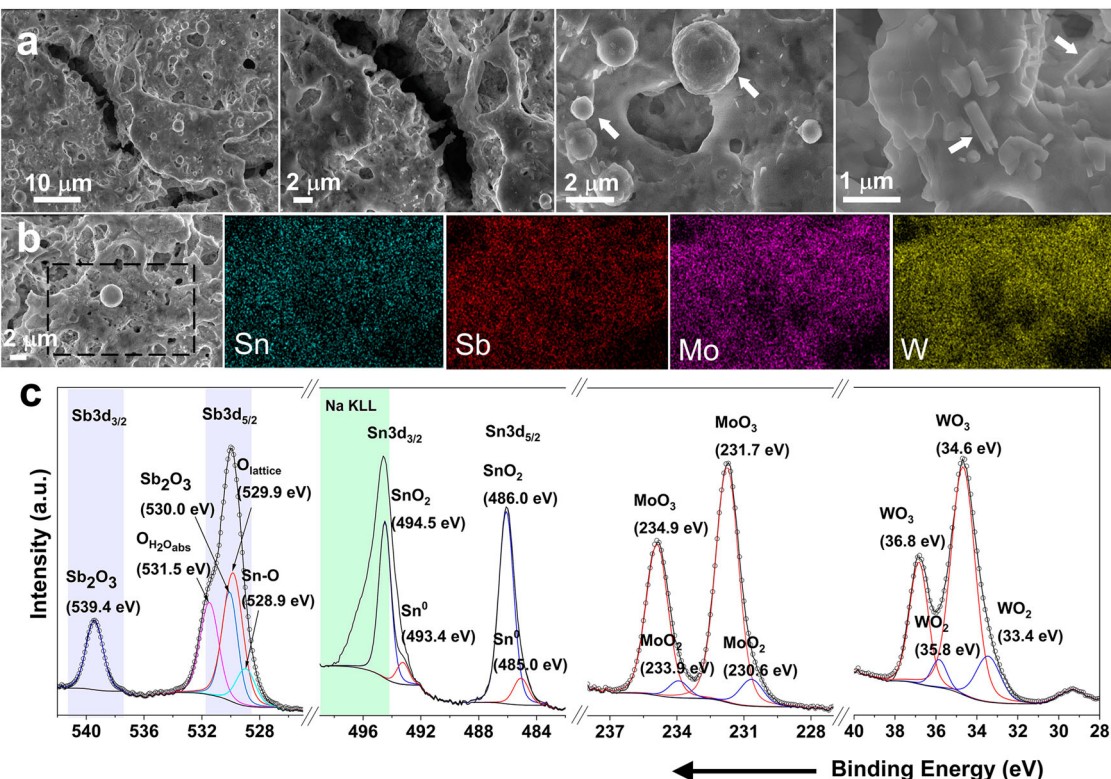

**Fig. 2 Morphology, elemental distribution, and valence states of the as-produced mixed oxide. a** SEM images and **b** EDX elemental mapping of the MO coating before the electrochemical measurements. The white arrows in (**a**) indicate spherical particles and rod-like structures. The dashed rectangle in (**b**) indicates the area where the EDX map was obtained. **c** High resolution XPS core-level spectra of Sb 3d/O 1s, Sn 3d, Mo 3d, and W 4 f. The black line represents the experimental data, and open circles correspond to the fitting.

scans, 5 mV s$^{-1}$, Fig. S3a) in the potential window of 1.2-1.8 V vs RHE. At pH = 2, the CVs showed a decreasing current density with each successive scan indicating a possible slow transpassive oxidation of the coating, ultimately leading to a stable material with nearly zero-current density in the studied region. At pH = 0, the first CV showed slightly higher current density and a more erratic behavior as compared to tests performed at pH = 1 and pH = 2. However, the subsequent CV scans mostly overlapped

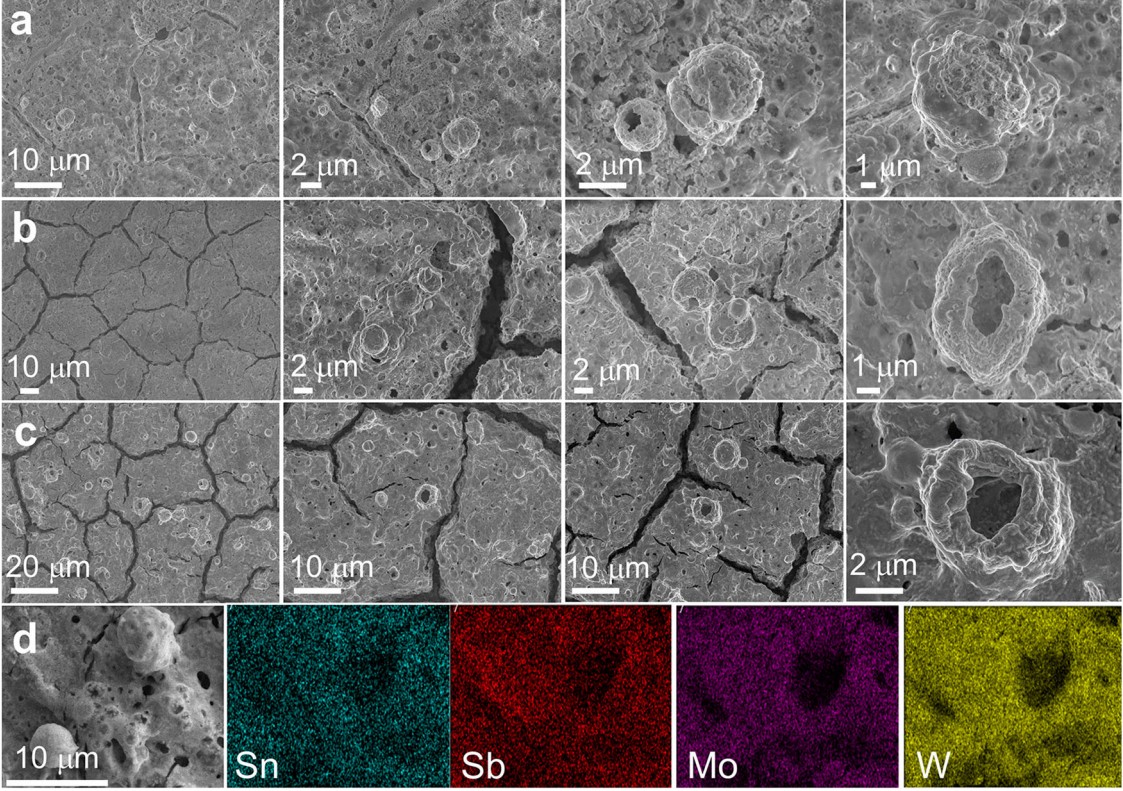

**Fig. 3 Changes in morphology after the electrochemical stress test at various pH.** SEM images after the electrochemical stress test of: (**a**) MO@pH2, (**b**) MO@pH1, and (**c**) MO@pH0; (**d**) EDX elemental mapping of MO@pH0.

indicating that the formed metal oxide layer was stable. This behavior can be rationalized when taking into consideration the XPS results in Fig. 2c, where low oxidation states of W, Mo and Sn were observed in the pristine MO coating. Indeed, the following set of chemical reactions can occur in the potential region of −0.2 to 0.2 V vs RHE and acidic conditions (pH ≤ 2)[30–32]:

$$MoO_2 + H_2O \rightarrow MoO_3 + 2H^+ + 2e^- \qquad (1)$$

$$WO_2 + H_2O \rightarrow W_2O_5 + 2H^+ + 2e^- \qquad (2)$$

$$W_2O_5 + H_2O \rightarrow WO_3 + 2H^+ + 2e^- \qquad (3)$$

$$Sn^{+0} + H_2O \rightarrow SnO + 2H^+ + 2e^- \qquad (4)$$

$$SnO + H_2O \rightarrow SnO_2 + 2H^+ + 2e^- \qquad (5)$$

Thus, in the selected pH range the first transpassivation reaction can be easily achieved at anodic potentials of 1.2 V vs RHE, leading to a fully oxidized MO coating where only the highest oxidation states of the metals are expected.

An additional stress test was later performed by using 500 CV scans at a higher scan rate of 100 mV s⁻¹ (see the voltammograms in Fig. S3b). Here, we aimed at forming a fully oxidized and stable coating for a subsequent material characterization. The fully oxidized coating was thoroughly characterized by SEM, EDX, and XPS. Figure 3a–c shows the SEM images of the MO coatings treated at different pH (0,1, and 2). An overall characteristic is the appearance of voids, fissures hollowed spheres, and tunnel-like structures through the entire coating. The severity of the defects increased as the pH of the electrolyte decreased as seen in Fig. 3(a-c). The appearance of these features might be related to the partial dissolution of metals, as discussed later. Such increase in porosity might lead to an increase in surface area, which is a desired characteristic when used as

catalyst support. In addition to these changes, the stressed-MO coatings displayed a considerable decrease in the XRD features (Figure S2b), indicating a drastic size reduction, or even amorphization of the nanocrystallites when compared to the pristine MO. It is also important to highlight that EDX elemental mapping (Fig. 3d) of the MO@pH0 sample still shows a homogeneous elemental composition through the entire surface despite the observed dissolution of material.

The atomic ratio of W/Mo and Sb/Sn was evaluated before and after the entire pH stress test to identify any deviation from the initial ratios (W/Mo = Sb/Sn = 1) caused by metal dissolution. The results from both EDX and XPS techniques are presented in Fig. 4. Clearly, the atomic ratio found by EDX and XPS exhibited similar trends. The discrepancy seen between XPS and EDX can be rationalized by considering that EDX has a larger penetration depth (~2 μm) than that of XPS (~6 nm), and under relatively mild pH conditions (e.g., pH = 2), the Mo dissolution might be more evident at the coating's top surface (first few nm) compared to the bulk parts of the coating. After the 500-CV test, the Sb/Sn ratio remained nearly constant along the studied pH range, indicating limited corrosion as expected from their theoretical Pourbaix diagrams[33,34]. However, as the pH of the electrolyte was decreased, the W/Mo ratio increased up to ~5.5 in both EDX and XPS analysis. Since WO₃ is stable under the selected conditions[35], the increased in W/Mo ratio indicates that Mo was dissolved during the electrochemical stress test as expected from its Pourbaix diagram[32]. The loss of Mo can explain the hollowed spheres and voids observed by SEM in Fig. 3a–c, as well as the current density observed in the voltammograms in Fig. S3. The W/Sb atomic ratio was also evaluated to corroborate that neither of W-Mo and Sb-Sn pairs were being dissolved simultaneously. The ratio W/Sb (Fig. S4) was found close to 3.5 after every electrochemical stress test, confirming that the content of W, Sb,

and Sn remained nearly constant under the studied pH conditions.

A more detailed analysis was conducted by performing the XPS peak-fitting for the MO@pH2, MO@pH1, and MO@pH0 coatings to identify changes in the metal oxidation states. Figure 5a summarizes the XPS results of W 4 f, Mo 3d, Sn 3d, and Sb 3d/O 1 s (overall spectrum presented in Fig. S5). As expected from the

reactions in Eqs. (1–5), all metals with low oxidation state were transformed into their high valence analogs, with various degrees of transformation according to the pH. For instance, tungsten was found solely in its higher valence $W^{+6}$ state after every test. In the case of Sn, at pH = 2 zero-valence Sn ($Sn^{+0}$) was transformed to SnO, but as the pH approached zero, all Sn was readily transformed into $Sn^{+4}$, whereas Sb exhibited a constant oxidation state ($Sb^{+3}$) related to $Sb_2O_3$. Regarding Mo, the Mo 3d peak-fitting revealed only the presence of $MoO_3$ at pH = 2 and 1, suggesting that an initial conversion of $Mo^{+4}$ to $Mo^{+6}$ had already occurred, and so Mo dissolution can be described with Eq. (7). However, at pH = 0 $MoO_2$ and $MoO_3$ were both detected indicating that Mo dissolution could have proceeded through Eqs. (6–7)[32] simultaneously.

$$MoO_2 + 2H_2O \rightarrow HMoO_{4(ac)}^{-1} + 3H^+ + 2e^- \qquad (6)$$

$$MoO_3 + H_2O \rightarrow HMoO_{4(ac)}^{-1} + H^+ \qquad (7)$$

It is also interesting to mention that in the Sn 3d region the contribution of Na KLL Auger electron (region highlighted in green) at pH = 2 was still detected, but at pH = 0 the Na signal disappeared. The loss of Na could also contribute to the formation of the hollowed structures seen in Fig. 3b, c. Finally, the O 1 s deconvolution revealed the peaks associated to lattice oxygen species ($O_{lattice}$)[28], low-coordinated defect sites ($O_{vac}$)[36], and adsorbed-water molecules ($O_{H2Oabs}$) (see Fig. S5)[29]. The newly $O_{vac}$ peak emerged after the electrochemical stress-test (absent in pristine MO coating), and by evaluating the $O_{vac}/O_{lattice}$, shown in Fig. 4, an increasing trend is observed when reducing the pH. This agrees with the Mo dissolution described by Eqs. (6–7), as well as the loss of Na. From these results, we conclude that the MO coating is moderately stable under the studied conditions, where after Mo and Na dissolution the resulting hollowed structures remain stable, this is exemplified in Fig. 5b.

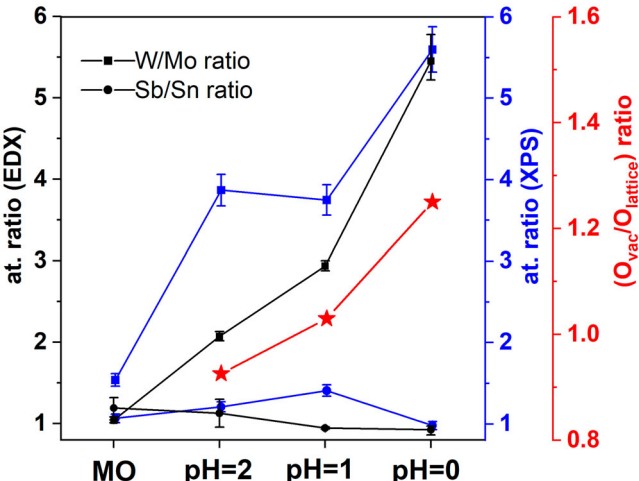

**Fig. 4 Changes in elemental composition after the electrochemical stress test at various pH.** W/Mo and Sb/Sn atomic ratio evaluated by EDX (black line), and XPS (blue line) for MO, MO@pH2, MO@pH1, and MO@pH0. The Ovac/Olattice ratio (red line) was obtained from the O 1 s high-resolution XPS core-level spectra in Fig. S5. The error bars represent the standard deviation of duplicate samples.

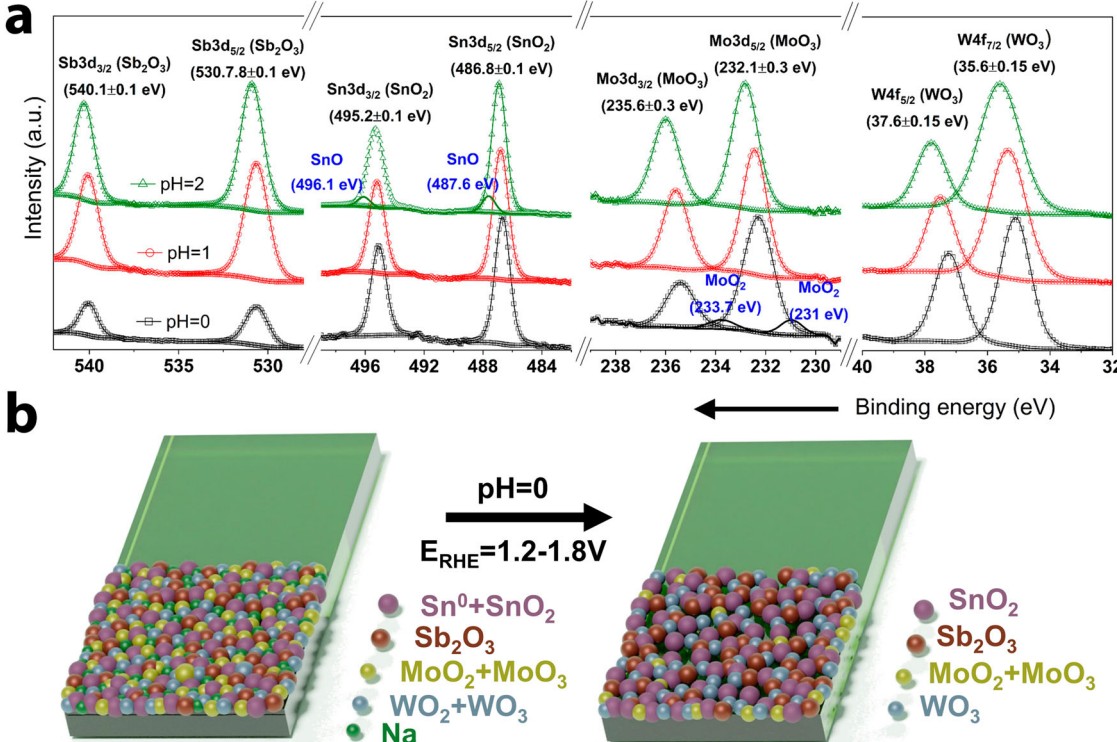

**Fig. 5 Changes in valence states and composition after the electrochemical stress test at pH = 0. a** High resolution XPS core-level spectra of Sb 3d/O 1 s, Sn 3d, Mo 3d, and W 4 f after the 500 CV test at different pH in the electrolyte, pH = 0 (open squares), pH = 1 (open circles), and pH = 2 (open triangles). **b** Schematic representation of the change in the MO composition after the electrochemical stress test at pH = 0.

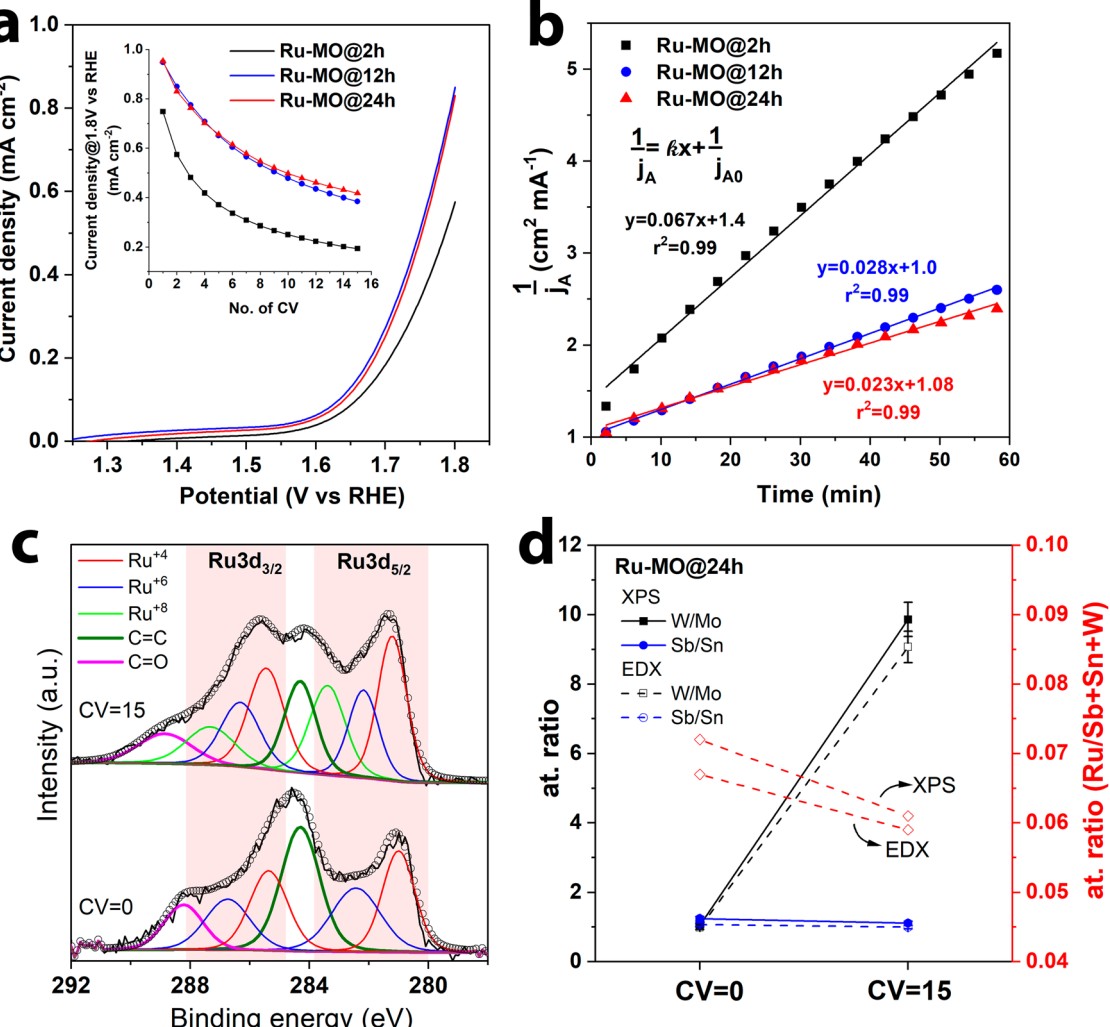

**Fig. 6 Electrocatalytic activity and stability of Ru-MO prepared at different annealing times. a** Polarization curves of Ru-MO@2 h, Ru-MO@12 h, and Ru-MO@24 h in 0.5 M H2SO4. The inset shows the change in current density evaluated at 1.8 V vs RHE through 15 CV scans. **b** Analysis of the kinetics of RuO2 deactivation. **c** High resolution XPS core-level spectra of Ru 3d for Ru-MO@24 h before and after 15 CVs. The black line represents the experimental data, and open circles correspond to the fitting. **d** W/Mo and Sb/Sn at. ratio and in the double y-axis the at. ratio of Ru with respect Sb, Sn, and W. The error bars represent the standard deviation of duplicate samples.

**Incorporation of Ru to MO**. After investigating and confirming the stability of the MO coating at different pH, in particular pH = 0, a small amount of Ru was anchored into the Sn-Sb-Mo-W mixture by adding RuCl3 in the precursor solution (2.8% w/w). Mo was kept as a part of the coating because its dissolution can be used to increase the surface area. This strategy has been reported in materials such as NiAlMo-based electrocatalysts where Al is leached to generate highly porous structures[37,38]. This also suggests that in our MO coatings, other sacrificial templates such as Ca or Al could be used instead of Mo. FTO glass was still used as a conductive substrate. At this stage, the annealing time (2, 12, and 24 h) was tuned to adjust the formation of the oxide network[39], while other parameters (temperature, molar ratio in the precursor solution, etc.) were kept unchanged. This new batch of samples were labeled as Ru-MO@2 h, Ru-MO@12 h, and Ru-MO@24 h. The polarization curves seen in Fig. 6a shows that both Ru-MO@12 h and Ru-MO@24 h exhibit the largest current density (~1 mA cm$^{-2}$ @ 1.8 V vs RHE) when compared to Ru-MO@2 h. From these results two key characteristics are worth noticing: (*i*) the low current density, and (*ii*) the large OER onset potential. These can be caused by the limited conductivity of the FTO glass, the low-Ru content in the electrode, and the substantially low conductivity of the Ru-MO coating. The sheet resistance of MO and Ru-MO@24 h are 12.2 MΩ sq$^{-1}$ and 1.5 MΩ sq$^{-1}$, respectively. The small amount of Ru incorporated into Ru-MO reduced nearly eight times the sheet resistance when compared to pristine MO. It has been reported that the addition of at least 20% w/w of noble metals to unary metal oxides is required to considerably improve both conductivity and OER activity[40,41]. This contrasts with the 2.8% w/w Ru content in the precursor solution used during the production of our electrodes. Therefore, the conductivity could be improved by increasing the amount of Ru, or by promoting the formation of conductive or small-band gap metal oxides. Nonetheless, at this stage we aim at studying the incorporation and stabilization of Ru into a matrix of stable oxides to produce a durable electrocatalyst for acidic media electrolysis, while the increase in conductivity without affecting electrochemical stability is a work in progress.

We tracked the degradation of the Ru-MO activity by monitoring the current density at 1.8 V vs RHE during a series of 15 CV scans. The results are plotted in the inset of Fig. 6a. From here, it can be concluded that longer annealing time improves the stability of the Ru-MO coating due to a better formation of the oxide network (e.g., low O$_{vac}$ content)[39]. Since

the current density at a given potential is proportional to the number of active sites for a given Ru-MO coating, we can evaluate the degradation rate by fitting the data (inset Fig. 6a) to the equation for a second-order reaction using:

$$\frac{1}{j_A} = kx + \frac{1}{j_{A0}} \tag{8}$$

where $k$ ($\text{cm}^2$ $(\text{mA·min})^{-1}$) is the rate constant associated to the deactivation of Ru, $j_A$ (mA cm$^{-2}$) is the current density evaluated at 1.8 V (vs RHE) in each successive CVs, $j_{A0}$ is the current density in the first CV, and $x$ is the time at which $j_A$ was recorded. The rate constant $k$ can be obtained from the slope of the linear fit shown in Fig. 6b, where smaller values of $k$ indicate greater Ru stability. Therefore, the observed stability trend is Ru-MO@24 h ($k = 0.023$ cm$^2$ $(\text{mA·min})^{-1}$) > Ru-MO@12 h ($k = 0.028$ cm$^2$ $(\text{mA·min})^{-1}$) > Ru-MO@2 h ($k = 0.067$ cm$^2$ $(\text{mA·min})^{-1}$).

The chemical composition of Ru-MO@24 before and after the 15 CV scans was investigated by high-resolution XPS and the peak-fitting is shown in Fig. S6. The analysis of pristine Ru-MO@24 (before any CV scan) revealed that Sn, Mo, and W already exhibited the highest oxidation state due to the longer annealing time used when compared to Ru-free MO. The high-resolution XPS of the Ru 3d overlaps with C 1s, so the parameters for the peak fitting were constrained by keeping the Ru 3d$_{5/2}$/Ru 3d$_{3/2}$ peak areas equal to 3:2, while also maintaining a spin-orbit splitting distance of 4.15 eV (regions highlighted in red)[42]. Then, the C = C and related bonds to carbon were added until the experimental data was properly fitted. The results for Ru-MO@24 h before the 15 CVs, Fig. 6c, reveal the existence of RuO$_2$ and RuO$_3$ at 280.9 eV and 282.4 eV, respectively[43,44]. After the 15 CVs, an additional peak at 283.4 eV attributed to the RuO$_4$ was observed[42,45]. Therefore, Ru dissolution proceeds through the transformation of RuO$_2$ to RuO$_4$ according to Eq. (9)[46]:

$$RuO_2 + 2H_2O \rightarrow RuO_{4(aq)} + 4H^+ + 4e^- \tag{9}$$

The W/Mo and Sb/Sn atomic ratios shown in Fig. 6d were evaluated using a similar approach as it was previously performed for pristine MO. The W/Mo and Sb/Sn ratios of Ru-Mo@24 h before the electrochemical test were as expected near to unity. After the 15 CVs scans, both EDX and XPS analysis show that the W/Mo ratio increases up to ~10, whereas the Sb/Sn remains nearly constant in agreement with the behavior seen in pristine MO. The Ru content was monitored relative to the metals that remained constant thought the stability test (Ru/(Sb+Sn+W)). As can be seen in Fig. 6d, the Ru at. ratio decreased after the test which correlates well with the observed catalysts deactivation, where RuO$_2$ is transformed into RuO$_4$ and subsequently dissolved to the electrolyte[30].

SEM studies were also performed on Ru-MO@24 h before (Fig. S7a) and after (Fig. S7c) the electrochemical stress test. SEM images of as-produced Ru-MO@24 h shows no major differences when compared to pristine MO, and EDX mapping (Fig. S7b) reveals a homogeneous distrubution of all metals. After the 15 CVs, the coating exhibited increased porosity, similar to the after-test MO samples, however this time it was originated by the loss of Mo, Na, and Ru, with the remaining material being moderately stable and with still homogeneous distribution of the elements (EDX mapping in Fig. S7d).

**Influence of the substrate**. After studying the performance and stability of Ru-MO deposited on FTO glass, we used titanium fiber felt as a substrate, which is a highly conductive and porous material commonly used as catalysts support and gas diffusion layer in PEM electrolyzers. The synthesis process was kept unchanged, and an annealing time of 24 h was used to ensure a fully oxidized coating as seen with Ru-MO@24 h. The new sample was labeled Ru-MO@Ti. The Ru content was estimated based on the material loading and EDX analysis, resulting in 2.64 % w/w of Ru introduced in the MO coating, in good agreement with the initial Ru concentration used in the precursor solution of 2.8% w/w Ru. The latter indicates that Ru-MO@Ti contains 110 µg of Ru cm$^{-2}$ of geometric area.

A reference sample containing solely Ru (similar Ru loading) was also deposited onto titanium fiber felt (labeled as RuO$_2$@Ti). A similar stability test of 15 CVs scans (1.2-1.8 V vs. RHE) at pH = 0 was performed. The results are presented in Fig. 7a where the second polarization curve is used to display only OER activity, since the first CV includes contribution from the transpassivation of titanium. As expected, RuO$_2$@Ti exhibited a higher current density and lower OER onset when compared to Ru-MO@Ti. This could be due to the direct contact between the substrate and RuO$_2$ catalyst. However, the activity of RuO$_2$@Ti drastically decreased to nearly 50% of the initial current density after only 15 CVs (see inset in Fig. 7a). On the other hand, the OER activity of Ru-MO@Ti improved in the first CV scans, probably due to an increase in surface area caused by Mo and Na dissolution, which in turn benefited electrolyte diffusion and Ru exposure. Note that the Ru-MO@Ti sample exhibited nearly 5 times higher current density at 1.8 V vs RHE than the Ru-MO@24 on FTO glass. The electrochemical surface area (ECSA) was evaluated for Ru-MO@Ti using the double layer capacitance method (see Fig. S8)[47], resulting in an ECSA of 131.4 cm$^2$. This value is comparable with other excellent supports used for noble metal electrocatalysts[40,48–51]. Afterwards, the mass activity (MA, A g$^{-1}$$_{Ru}$) was calculated using Eq. (10)[52,53],

$$MA = \frac{j_k}{m_{Ru}} \tag{10}$$

where $j_k$, and $m_{Ru}$ denote the kinetic current density (A cm$^{-2}$), and Ru loading density (g$^{-1}$$_{Ru}$ cm$^{-2}$$_{ECSA}$), respectively. We obtained a $MA$ of 4440 A g$^{-1}$$_{Ru}$, 1620 A g$^{-1}$$_{Ru}$, and 985 A g$^{-1}$$_{Ru}$ at 1.80, 1.70, 1.65 V (vs RHE), a comparable value to other recent OER electrocatalysts (600 A g$^{-1}$$_{@1.48V(RHE)}$[54], 3760 A g$^{-1}$$_{@1.55V(RHE)}$[49], 380 A g$^{-1}$$_{@1.53V(RHE)}$[55]).

The analysis of the Tafel slopes (Fig. 7b) indicates that for RuO$_2$@Ti (115 mV dec$^{-1}$) the water adsorption towards the active site is the rate determining step (rds), as described in Eq. (11)[56–58]:

$$S + H_2O \rightarrow S - OH^* + H^+ + e^- \tag{11}$$

where $S$, and $S$-$OH^*$ denote the active site and adsorbed intermediate species, respectively. The high Tafel slope of RuO$_2$@Ti can be due to the low catalyst loading. It has been reported that a Tafel slope in the range of 30-40 mV dec$^{-1}$ is obtained with RuO$_2$ loadings within 20 to 30% w/w[57–59], having instead, the second electron transfer as the rds, as denoted in Eq. (12).

$$S - OH^* \rightarrow S - O_{ads} + H^+ + e^- \tag{12}$$

Likewise, the Tafel slope of Ru-MO@Ti was equal to 234 mV dec$^{-1}$, such a large value is still associated to the water adsorption being the rds, similar to RuO$_2$@Ti, and likely caused by the low catalyst loading.

Electrochemical impedance spectroscopy (1.7 V vs RHE, Fig. S9) is used to evaluate the ohmic (R$_\Omega$) and charge transfer (R$_{CT}$) resistances. Both materials exhibited similar R$_\Omega$ values (Table S1), however a larger R$_{CT}$ was observed for Ru-MO@Ti, likely due to the low conductivity of the MO protective scaffold that restricted the transport of the electrons to the active sites.

The stability test was further evaluated by a longer electro-chemical test on new pieces of Ru-MO@Ti and RuO$_2$@Ti

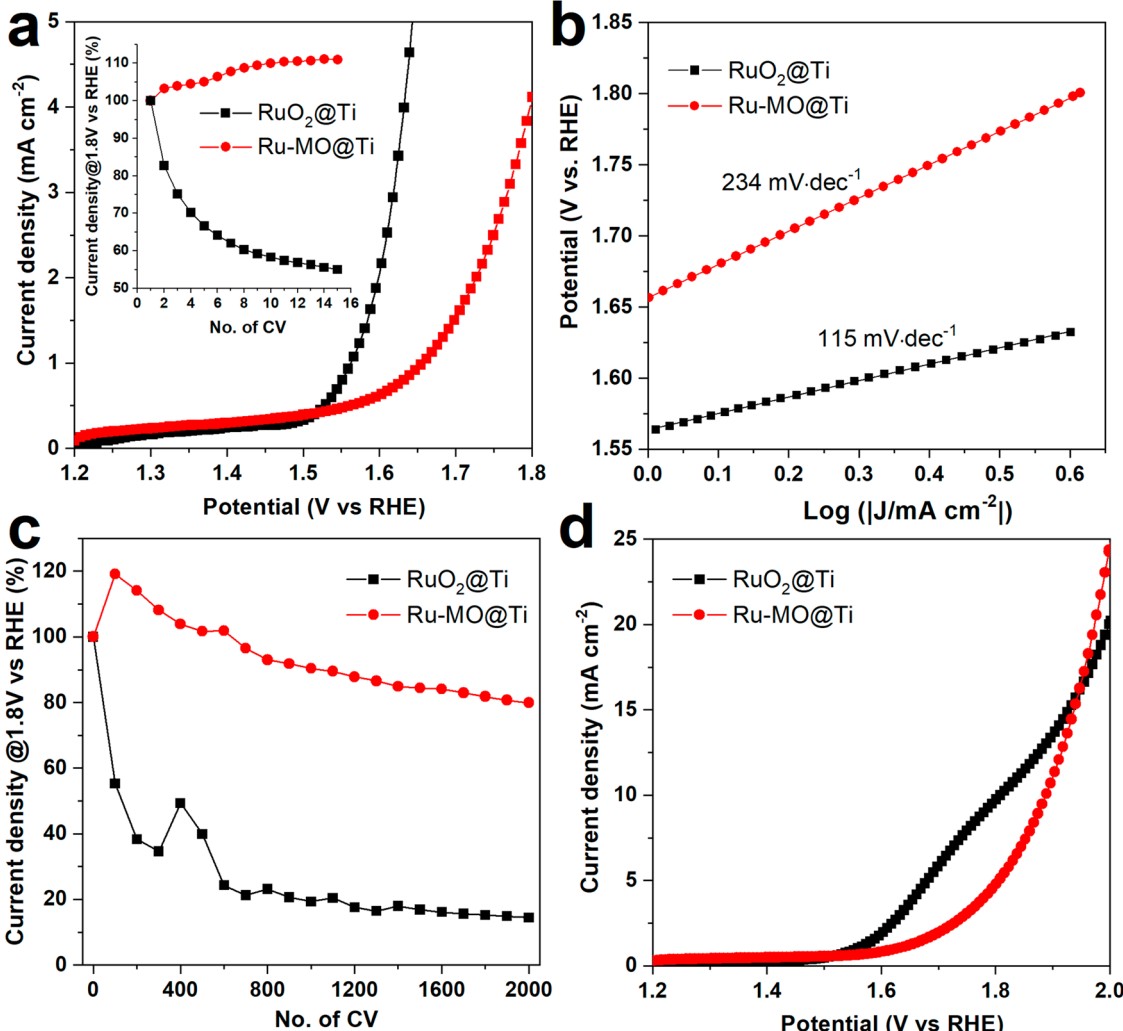

**Fig. 7 Electrocatalytic activity and stability of Ru-MO on titanium felt. a** Polarization curve of Ru-MO@Ti in 0.5 M H2SO4. (Inset) Relative current density at 1.8 V vs RHE recorded during 15 CV scans. **b** Tafel slopes of Ru-MO@Ti, and RuO2@Ti. **c** 2000-CVs stability test. The relative current density at 1.8 V vs RHE was recorded every 100 CVs. **d** Polarization curves taken after the 2000-CVs test.

samples. This time the test consisted of 2000 CVs at a high scan rate in which we monitored the current density at 1.8 V vs RHE by performing low scan-rate LSVs after every 100 CVs. This current density was then expressed relative to the first LSV, and the results were plotted in Fig. 7c. As expected, the current density of Ru-MO@Ti increases above 100% in the first CVs due to the increase in surface area by the partial dissolution of Mo and Na. But more interestingly, after the 2000 CVs the current density of Ru-MO@Ti was only reduced by 20%, whereas RuO2@Ti exhibited a much faster degradation ( ~ 50% after 100 CVs) with a decrease in current density of nearly 90% after the 2000 CVs. These results confirm that the produced mixed oxide provides stability to Ru while still being active towards the OER in harsh acidic conditions. Following the stability test, a final polarization curve was recorded in the potential window of 1.2-2 V vs RHE. In Fig. 7d, the RuO2@Ti exhibits an additional passivation between 1.7 and 1.9 V vs RHE, contrary to the Ru-MO@Ti coating where no additional features are observed. These results indicate that the MO coating protects both the Ru and the Ti substrate.

Similar conclusions are obtained when performing a chronopotentiometry study (2.5 mA cm$^{-2}$ for 10 h, see Fig. S10) instead of the 2000-CV test. After 10 h of operation, Ru-MO@Ti experienced only a minor increase in potential of 10 mV, while RuO2@Ti exhibited a substantial degradation with an increase of

500 mV after 10 h. These results highlight the importance of the Sn-Sb-Mo-W mixed-oxide as an effective and protective environment for Ru, reducing its corrosion and extended the operational stability during OER in acidic media. It should also be noticed that the developed coating avoided degradation of the titanium fiber felt since no further passivation was observed in Ru-MO@Ti. Therefore, the MO coating can be used to protect both Ru electrocatalyst and Ti gas diffusion layer, extending their lifetime when used in PEM electrolyzers.

Finally, both Ru-MO@Ti and RuO2@Ti were characterized by SEM and XPS before and after the 2000-CVs stability test. The as-produced Ru-MO@Ti exhibited a similar morphology than the one observed when using FTO glass as substrate (Fig. 8a, b). On the other hand, RuO2@Ti had a thin and inhomogeneous coating (Fig. S11a) with the occasionally micrometer sphere and rod structures. After the stability test, Ru-MO@Ti presented larger fissures, and apparently more exposed Ti substrate (Fig. 8c, d). In the case of RuO2@Ti, the lack of a protective MO environment resulted in a notably exposed Ti surface due to a clear RuO2 dissolution (Fig. S11b), in agreement with the decrease in current density seen in Fig. 7c. XPS analysis before the 2000-CV test (Fig. 8e) indicate that Ru-MO@Ti only contains RuO2 (280.4 eV) and RuO3 (281.8 eV)[43,44]. In contrast, RuO2@Ti exhibited peaks corresponding to RuO2 and its satellites (280.7/282.7 eV), RuO3

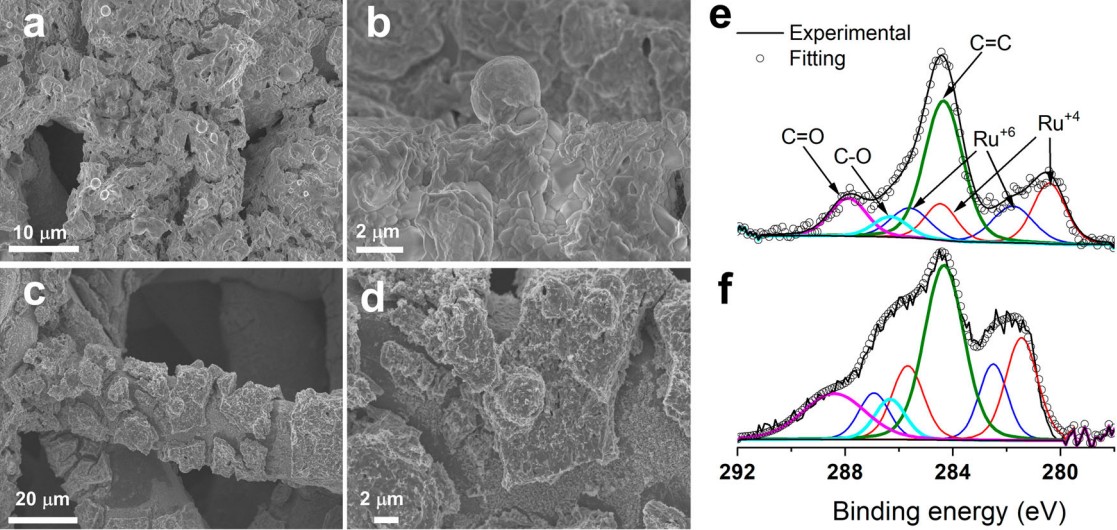

**Fig. 8 Changes in morphology and ruthenium valence states of Ru-MO after the stability test. a**, **b** SEM images of Ru-MO@Ti before the 2000-CVs stability test. **c**, **d** SEM images of Ru-MO@Ti after the stability test. **e**, **f** High-resolution XPS core-level spectra of Ru 3d for Ru-MO@24 h before and after the stability test, respectively.

(281.6 eV), and $RuO_4$ (283.7 eV) in samples characterized before and after the stability test (see Fig. S11c, 1d)[42,44]. The existence of higher valence ($Ru^{8+}$) in a solid has been previously observed during the transformation of Ru precursors into their respective oxides[45]. The presence of $RuO_4$ can explain the fast degradation seen during the stability test. Remarkably, after the stability test Ru-MO@Ti did not show any $RuO_4$ contribution (Fig. 8f), on the contrary the signals from both $RuO_2$ and $RuO_3$ increased when compared to the as-prepared material, confirming that the dissolution of Mo and Na exposed additional Ru.

## Conclusions

We investigated the production of a quaternary mixed-oxide matrix suitable for stabilizing and protecting both $RuO_2$ and Ti substrate in acidic environments under anodic potentials. The acid-stable oxide was produced as a coating using solution precursor plasma spraying creating a well-interconnected network of nanostructured $WO_3$, $Sb_2O_3$, $SnO_2$, and $MoO_3$. The role of these oxides is to create a stable scaffold where Ru can be anchored, improving its resistance against corrosion under anodic potentials and acidic environments, the latter is possible due to impeded formation of $O_{vac}$ as well as higher valence ruthenium ($Ru^{+8}$). The Sb-Sn-Mo-W mixed-oxide coating deposited on FTO glass showed indeed excellent stability in a wide range of acidic conditions (from pH = 2 to 0) finding that only $MoO_x$ was partially dissolved leading to a porous structure. This effect may also be achieved by using other inexpensive elements such as Ca or Al. Low content of $RuO_2$ (2.64% w/w Ru) was successfully anchored into the mixed-oxide matrix leading to enhanced stability during OER at low pH when deposited on titanium fiber felt. The superior stability of the Ru was attributed to the stabilization of the low oxidation states ($Ru^{2+}$ and $Ru^{3+}$), while suppressing the formation of $RuO_4$. Besides, the catalyst supported on titanium fiber felt led to an ECSA of 131.4 cm$^{-2}$ associated to a mass activity of 4440 A g$^{-1}_{Ru}$ at 1.8 V (vs RHE), a highly competitive value compared to other Ru-based state-of-the-art electrocatalysts. Our approach also opens the possibility to stabilize other inexpensive OER active metals, such as Fe, Ni, or Co, inside the MO matrix. Finally, we have shown that multimetal oxides have the potential to extend the lifetime of $RuO_2$ and Ti-based gas diffusion layers during OER in acidic media, which is one of the main limitations for industrial-scale renewable hydrogen production.

## Methods

**Materials and reagents**. Tin (IV) chloride pentahydrate ($SnCl_4 \cdot 5H_2O$, 98%), sodium tungstate dihydrate ($Na_2WO_4 \cdot 2H_2O$, ≥99%), ammonium molybdate tetrahydrate (($NH_4$)$_6Mo_7O_{24} \cdot 4H_2O$, ≥99%), antimony chloride ($SbCl_3$, ≥99%), ruthenium chloride hydrate ($RuCl_3 \cdot xH_2O$, ruthenium content about 40-49%), sulfuric acid ($H_2SO_4$, 95-97%) and phosphoric acid ($H_3PO_4$, 85%) were acquired from Merck (Sigma-Aldrich). $RuCl_3 \cdot xH_2O$ was kept under vacuum to avoid water absorption. Disodium hydrogen phosphate ($Na_2HPO4$, 99%) and potassium dihydrogenphosphate ($KH_2PO_4$, 99.5%) were purchased from VWR. All chemicals were used as received. Fluorine doped tin oxide (FTO) coated glass slides (7 Ω sq$^{-1}$) were acquired from Merck (Sigma-Aldrich). Titanium fiber felt (0.2–0.3 mm thickness, 53-56% porosity) was purchased from Fuel Cell Store.

**Material characterization**. SEM studies were carried out on a Carl Zeiss Merlin microscope equipped with an energy dispersive X-ray spectrometer. XPS was performed on a Kratos Axis Ultra DLD electron-spectrometer equipped with a monochromatic X-ray source (Al Kα line of 1486.6 eV). X-ray diffraction (XRD) characterization was conducted on a PANalytical X'pert diffractometer ($\lambda = 1.5406$ Å, Cu Kα) in the range of 5 to 75 degrees (step size of 0.01395° and time of 0.5 s step$^{-1}$) at atmospheric conditions. Sheet resistance was measured in an Ossila Four-Point Probe System. These measurements were performed in a cleanroom with 52% controlled humidity and temperature (21 °C).

**Production of mixed oxides**. Mixed oxide coatings based on Sn-Sb-Mo-W were deposited onto FTO glass substrates using SPPS. Before use, FTO substrates (1 × 2 cm) were thoroughly cleaned in EtOH/$H_2O$ (70% v/v) by sonication for 1 h, then rinsed with water (18.2 MΩ·cm). An aqueous precursor solution containing Sn, Sb, Mo, and W with molar ratio 5:5:18:18 and a metal concentration of 0.1 M was used as liquid feedstock during the SPPS process. The metal concentration was selected to have the highest metal content without causing precipitation that could lead to heterogeneous solution and coating. An atmospheric plasma spraying system (Metallisation Met-PCC (PLAS)) equipped with a PL50 pistol and a 6 mm nozzle was used. A mixture of argon (50.0 NL min$^{-1}$) and nitrogen (2.0 NL min$^{-1}$) was employed as medium to create the plasma while applying 500 A of current, resulting in a power output of 25.5 kW. The solution precursor (flow rate of 15 mL min$^{-1}$) was pre-mixed with nitrogen gas (3 NL min$^{-1}$) to create a fine mist before the injection into the plasma plume. The plasma torch was controlled by a robotic arm (ABB 2600) with a raster spraying pattern with a lateral velocity of 250 mm s$^{-1}$ and a vertical displacement of 4 mm per turn forming one cycle; a total of eight cycles were used to form one layer in a total area of ~180 cm$^2$. Several FTO substrates were placed in the center of the spraying area where a mask limited the coated area to 1 × 1 cm$^2$ in each substrate. A total of 10 substrates were coated simultaneously. The distance between the substrates and the plasma torch was kept constant to 200 mm. A complete coating was achieved by applying a total of 16 layers. After the spraying process, the coated FTO substrates were oxidized in air using an oven at 500 °C for 2 h with a ramp of temperature of 4.3 °C min$^{-1}$. This temperature was selected to maximize the formation of the oxide network without compromising the stability of the FTO glass substrate. The resulting mixed oxide coatings were labeled as MO. The complete process is depicted in Fig. 1. Finally, the MO samples were cooled down and stored at ambient conditions.

**Production of MO with RuO₂.** The MO coating was used as a protective environment to stabilize ruthenium as active metal for OER. Here, 0.25 g of $RuCl_3 \cdot xH_2O$ were added into the precursor solution (100 ml) leading to a Ru:Sn:Sb:Mo:W molar ratio of 1.7:5:5:18:18, and a metal concentration of 0.112 M. The added Ru mass precursor resulted in a 2.8% w/w Ru content in the solution precursor. The Ru-containing coating was deposited onto FTO substrates (10 pieces simultaneously) following the same spraying and annealing procedures used to generate the MO coatings. To assess the stability of the Ru in the coating, the length of the oxidation process was tuned for 2 h, 12 h and 24 h. These samples were labeled as Ru-MO@2 h, Ru-MO@12 h, and Ru-MO@24 h, respectively.

Additional samples of Ru-MO were deposited onto titanium fiber felt following the same procedure as in the FTO substrate. The oxidation process was performed at 500 °C for 24 h. The sample was labeled as Ru-MO@Ti. Lastly, a reference sample comprising only RuO₂ (0.25 g of $RuCl_3 \cdot xH_2O$ in 100 ml of DI water) was produced on titanium fiber felt by using a solution precursor containing solely $RuCl_3 \cdot xH_2O$, (labeled as RuO₂@Ti). This sample was used for comparison and was subjected to the same oxidation process as the Ru-MO@Ti.

**Electrochemical measurements.** The electrochemical measurements were performed in a conventional three-electrode cell at room temperature. All the data were collected in an Autolab electrochemical workstation. The MO and Ru-MO samples were used as working electrodes, whereas a Pt wire was selected as counter electrode, and an Ag/AgCl (3 M KCl) electrode as a reference electrode. All the measured potentials reported in this work were converted to the reversible hydrogen electrode (RHE) using the Nernst equation ($E_{RHE} = E_{Cell} + 0.059$ pH + $E_{Ag/AgCl}$) $E_{Ag/AgCl} = 0.210$ V vs. SHE, and $E_{Cell}$ is the experimental measured potential. A series of two stress test were conducted to evaluate the stability of MO electrodes at different pH (0, 1 and 2). The first test consisted in 20 cyclic voltammetry (CV) scans between 1.2 and 1.8 V vs RHE with 5 mV s$^{-1}$ scan rate, followed by 500 CVs measured in the same potential range but using 100 mV s$^{-1}$ as scan rate (samples labeled as MO@pH0, MO@pH1 and MO@pH2). The electrolyte solutions with pH = 0 and pH = 1 were prepared using H₂SO₄ with concentrations of 0.5 M and 0.05 M, respectively. For the electrolyte at pH = 2, a buffer solution was prepared dissolving 8.95 g of Na₂HPO₄ and 3.4 g of KH₂PO₄ in 1 L of DI water, and the pH was adjusted adding enough H₃PO₄. All the electrolyte solutions were bubbled with high purity argon for 30 min prior to the experiments. The activity of Ru-MO, Ru-MO@Ti and RuO₂@Ti samples were assessed by monitoring the OER activity using 15 CVs in the range of 1.2 and 1.8 V vs RHE with 5 mV s$^{-1}$ scan rate at pH = 0. The electrochemical surface area (ECSA) was evaluated by using the double layer capacitance ($C_{dl}$) method by recording 10 CVs in the non-faradaic region (1.3-1.4 V vs RHE) at various scan rates. A value of 35 $\mu$F cm$^{-2}$ was used for the specific capacitance[47]. For the Ru-MO@Ti and RuO₂@Ti samples, the stability was evaluated by measuring 2000 CVs in the range of 1.2 and 1.8 V vs RHE with a scan rate of 100 mV s$^{-1}$, but every 100 CVs a single sweep voltammetry (5 mV s$^{-1}$) was recorded. Finally, one LSV was taken in the range of 1.2 and 2 V vs RHE. Electrochemical impedance spectroscopy (EIS) studies were carried out in potentiostat Ivium Technologies by applying a potential of 1.7 V vs RHE in a frequency range of 1000 Hz to 0.1 Hz (10 mV amplitude).

**Statistics and reproducibility.** Data obtained from SEM and EDX is presented without alterations in contrasts or brightness. XRD data is plotted without alterations and indexed using the HighScore software version 4.5. The XPS data was fitted using CasaXPS version 2.3.24PR1.0. The EIS analysis was performed using the software Ivium Technologies version 4.1.

## Data availability

All data generated during and/or analyzed during this study are included in this published article and its supplementary information files.

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

## Acknowledgements

E.G.-E. acknowledges support from Vetenskapsrådet (2018-03937), the Kempe Foundation (JCK-2132), and the Carl Tryggers Foundation (CTS 21-1581). We also thank the Umeå Core Facility for Electron Microscopy (UCEM), the Vibrational Spectroscopy Core Facility (ViSp), the XPS platform at Umeå University.

## Author contributions

A.P.-G. designed and performed the experiments, contributed to data analysis, and writing. X.W. contributed to data analysis. M.R. contribute to data analysis. P.J.M. performed the experiments. E.G.-E. conceived and supervised the project, contributed to data analysis, and writing.

## Funding

## Competing interests

The authors declare no competing interests.
