## [Peer Review file · Communications Engineering]

A Quaternary Mixed Oxide Protective Scaffold for Ruthenium During Oxygen Evolution Reaction in Acidic MediaReviewers' comments:

Reviewer #1 (Remarks to the Author):

This paper reports new quaternary Sn-Sb-Mo-W mixed oxide scaffold for Ru prepared by solution precursor plasma spraying for oxygen evolution reaction in acidic media. Physical and electrochemical characteristics of Ru-Mo@Ti were systematically studied and compared with those of Ru@Ti. Accelerated stress test of two specimens was also conducted under severe potential cycling between 1.2 and 1.8 V vs. RHE. This paper is well-written, and this work is worthy of subject to be published in this journal. Nonetheless, it should be considered several comments below to further improve this paper.

1. Provide the reason or experimental evidence why the authors selected i) the annealing temperature for the electrode to be 500 °C and ii) the molar ratio of electrode materials to be 1.7:5:5:18:18, iii) metal concentration to be 0.1M in this study.
2. Clarify why in figure 4, W/Mo atomic ratio for MO@pH2 determined by XPS is relatively high, compared to that determined by EDX and even higher than that for MO@pH=1 by XPS.
3. If Mo was used as sacrificial materials by dissolution from the electrode, it may be replaced by one of inexpensive materials. Is there any reason to stick to Mo?
4. Describe how to obtain the kinetic current density, j_k . In this study, mass activity of Ru@Ti and Ru-MO@Ti was determined at 1.8 V vs. RHE where mass transfer effect was not negligible even with iR-correction because open circuit potential was assumed to be much lower than 1.8 V vs. RHE.
5. At line 430, "Likewise, the Tafel slope of Ru-MO@Ti was equal to 234 mV dec⁻¹ with similar rds as RuO₂@Ti" Tafel slope of RuO₂@Ti is indeed a half of that of Ru-MO@Ti. Clarify this sentence.

Reviewer #2 (Remarks to the Author):

In this paper, the authors investigated the synthesis of Sb-Sn-Mo-W mixed-oxide stabilized RuO₂ and studied the OER performance in acidic environments. The mixed oxides can improve the stability of RuO₂ in harsh reaction environments. However, the enhanced stability along with significantly decreased the catalytic activity and increased overpotentials of RuO₂, as shown in Figure 7. Generally, the extended stability may always occur when the catalytic activity is low. Meanwhile, in the paper, the authors did not investigate why these kinds of metal oxides were chosen and how they can promote the stability of RuO₂. As such, I cannot recommend it can be published by Communications Engineering.

Reviewer #3 (Remarks to the Author):

In this manuscript, the authors reported the quaternary Sn-Sb-Mo-W mixed oxide as protective scaffold for oxygen evolution active metals in acidic media. The findings from this work are interesting and meaningful. I suggest accepting this work after major revisions.

1. A reduction of 20% in OER activity after 2000 CVs is not satisfied compared to the reported values.
2. The roles of WO₃, Sb₂O₃, SnO₂, and MoO₃ should be further highlighted.
3. What about the solubility of WO₃, Sb₂O₃, SnO₂, and MoO₃ during the acid OER process?
4. The reported current density is low and how to fulfill the requirements of large-scale application of PEM water electrolysis?
5. The following papers (Nature Nanotechnology, 2021 16, 1371–1377; Molecules, 2021, 26, 18, 5476; Energy & Environmental Materials, 2022, e12441) are recommended to be cited for improving the manuscript.

We appreciate the constructive comments and suggestions brought by all reviewers. We have addressed all their concerns in the point-by-point list. All changes made to the manuscript are also included, and a revised version of the manuscript with highlighted changes is included in the submission files. With all these, we hope that the manuscript can be now accepted for publication.

Reviewer #1

This paper reports new quaternary Sn-Sb-Mo-W mixed oxide scaffold for Ru prepared by solution precursor plasma spraying for oxygen evolution reaction in acidic media. Physical and electrochemical characteristics of Ru-Mo@Ti were systematically studied and compared with those of Ru@Ti. Accelerated stress test of two specimens was also conducted under severe potential cycling between 1.2 and 1.8 V vs. RHE. This paper is well-written, and this work is worthy of subject to be published in this journal. Nonetheless, it should be considered several comments below to further improve this paper.

Comment #1: Provide the reason or experimental evidence why the authors selected i) the annealing temperature for the electrode to be 500 °C and ii) the molar ratio of electrode materials to be 1.7:5:5:18:18, iii) metal concentration to be 0.1M in this study.

Answer: All pre-defined parameters were selected after various optimization processes. We initially did not include many details about them due to space constrains and focus of the manuscript. But now, we have included a brief explanation of the origin of such parameters. Here is some in-depth explanation:

(i) The quaternary mixture contains metals that can have various levels of oxidation states. So, we aimed at ensuring a complete oxidation of the scaffold with all metals in their highest oxidation state. We tested various temperatures, but 500 °C was the maximum without compromising the stability of the FTO glass substrate. However, despite using 500 °C we still detected an incomplete conversion of metal precursors to their respective oxides (see **Figure 2c** in main manuscript). This part of the study was not included in the manuscript, but now we have added this information in the Methods section as seen below:

“After the spraying process, the coated FTO substrates were oxidized in air using an oven at 500 °C for 2 h with a ramp of temperature of 4.3 °C per min. This temperature was selected to maximize the formation of the oxide network without compromising the stability of the FTO glass substrate. The resulting mixed oxide coatings were labelled as MO.”

(ii) The molar ratio of 5:5:18:18 for Sn:Sb:Mo:W was selected after some preliminary experiments indicating that binary metallic oxides of WO_x and MoO_x (W:Mo = 1:1 molar ratio), SnO_2 and Sb_2O_3 (Sn:Sb = 1:1 molar ratio), as well as WO_x and SnO_2 (W:Sn = 18:1 molar ratio) can form stable oxides. These preliminary results are briefly mentioned in the Result and Discussion section, and the data is shown in **Figure S1**.

Paragraph already in the manuscript: “The molar ratio of the metal precursors Sn:Sb:Mo:W was set to 5:5:18:18. This particular ratio was selected based on a previous theoretical report,^{14,17,18} and our own preliminary experimental studies (**Figure S1**), both indicating that binary metallic oxides of WO_x and MoO_x (W:Mo = 1:1 molar ratio), SnO_2 and Sb_2O_3 (Sn:Sb = 1:1 molar ratio), as well as WO_x and SnO_2 (W:Sn = 18:1 molar ratio) can form stable mixed oxides under acidic and anodic potentials.”

Regarding the Ru molar ratio. The Ru concentration was the minimum to significantly trigger the OER. As it was mentioned in the manuscript, a loading of 110 μg of Ru per cm^2 of geometric area was achieved when a solution precursor with molar ratio 1.7:5:5:18:18 was employed under the selected synthesis conditions. Lower Ru concentration in the solution leads to traces amount of Ru in the substrate with minimum effects in activity.

No changes were made to the manuscript.

(iii) To ensure reproducibility and reduce inconsistency between different sample batches, the solution precursor must not exhibit significant changes in short periods of time. Therefore, the concentration in the solution was fixed at 0.1 M because higher concentrations lead to precipitation before the spraying process. We have added this information to the experimental section:

“An aqueous precursor solution containing Sn, Sb, Mo, and W with molar ratio 5:5:18:18 and a metal concentration of 0.1 M was used as liquid feedstock during the SPPS process. The metal concentration was selected to have the highest metal content without causing precipitation that could lead to heterogeneous solution and coating.”

Comment #2. Clarify why in figure 4, W/Mo atomic ratio for MO@pH2 determined by XPS is relatively high, compared to that determined by EDX and even higher than that for MO@pH=1 by XPS.

Answer: XPS is a surface technique with a relatively small penetration depth of ~6 nm and a spot size of 10-200 μm . While EDX has a typical depth analysis of 2 μm and much larger spot size. Therefore, some discrepancies can be expected between the surface composition seen by XPS and the semi-bulk analysis of EDX.

Regarding the disagreement with the W/Mo ratio. The W/Mo ratio seen at pH = 2 by XPS is larger than that of EDX because Mo has been primarily removed from the top surface of the coating. While, EDX is sampling parts of the bulk where Mo is present in larger quantities, meaning that a much smaller W/Mo ratio is observed. All these indicates out that at a lesser acidic condition (e.g. pH = 2), the loss of Mo at the top surface is significantly larger than that at the bulk. While at lower pH conditions the amount of Mo removed from the top surface and the bulk (up to ~2 μm) after the electrochemical study is similar, and thus smaller discrepancies are observed.

We have added a brief discussion about this deviation in the main text:

“The results from both EDX and XPS techniques are presented in **Figure 4**. Clearly, the atomic ratio found by EDX and XPS exhibited similar trends. The discrepancy seen between XPS and EDX can be rationalized by considering that EDX has a larger penetration depth (~2 μm) than that of XPS (~6 nm), and under relatively mild pH conditions (e.g. pH = 2), the Mo dissolution might be more evident at the coating’s top surface (first few nm) compared to the bulk parts of the coating. After the 500-CV test, the Sb/Sn ratio remained nearly constant along the studied pH range...”

Comment #3. If Mo was used as sacrificial materials by dissolution from the electrode, it may be replaced by one of inexpensive materials. Is there any reason to stick to Mo?

Answer: We agree with the reviewer that Mo could be replaced by another cheaper element such as Ca or Al. However, we believe that in-depth studies should be carried out to identify the changes/effects (morphology, crystal structure, stability, etc.) of replacing Mo. All these could be investigated in future studies. We have added a brief comment about this possibility as seen below:

Discussion section: “Mo was kept as a part of the coating because its dissolution can be used to increase the surface area. This strategy has been reported in materials such as NiAlMo-based electrocatalysts where Al is leached to generate highly porous structures. This also suggests that in our MO coatings, other sacrificial templates such as Ca or Al could be used instead of Mo. FTO glass was still used as a conductive substrate.”

Conclusions: “The Sb-Sn-Mo-W mixed-oxide coating deposited on FTO showed excellent stability in a wide range of acidic conditions (from pH=2 to 0) finding that only MoO_x was partially dissolved leading to a porous structure. This effect may also be achieved by using other inexpensive elements such as Ca or Al.”

Comment #4: Describe how to obtain the kinetic current density, j_k . In this study, mass activity of Ru@Ti and Ru-MO@Ti was determined at 1.8 V vs. RHE where mass transfer effect was not negligible even with iR-correction because open circuit potential was assumed to be much lower than 1.8 V vs. RHE.

Answer: Mass transfer effects were neglected since they become significant at large current densities (e.g., 200 mA cm⁻²) [Liu, P. et al. Adv.Mater.2021, 33, 2007377, Luo, Y., et al Nat Commun, 10, 269, 2019]. In addition, impedance spectroscopy studies at 1.7 V vs RHE (**Figure S9**) indicate that mass transport limitations are not relevant due to the absence of features at low frequency regions, either a semi-circle indicating a finite mass transport behaviour (e.g., process under mechanical agitation), or a linear diffusion feature related to a semi-infinite Warburg diffusion (e.g., process without mechanical agitation) (J. Phys. Chem. C 2019, 123, 35, 21440–21447, RSC Adv. 2020 Aug 17; 10(51): 30887–30895.). Moreover, the polarization curve at higher potentials (**Figure 7d** in main manuscript (red line)) of Ru-

MO@Ti does not exhibit any change in the slope after OER is initiated. Therefore, we conclude that under the selected conditions, no limitation due to mass transfer are observed at 1.8 V vs RHE.

Comment #5. At line 430, ‘Likewise, the Tafel slope of Ru-MO@Ti was equal to 234 mV dec⁻¹ with similar rds as RuO₂@Ti’ Tafel slope of RuO₂@Ti is indeed a half of that of Ru-MO@Ti. Clarify this sentence.

Answer: The large Tafel slope seen on both systems exceeds the theoretical value of 120 mV dec⁻¹ corresponding to water adsorption onto the active site. Therefore, values of 120 mV dec⁻¹ or larger are associated to water adsorption being the rate determining step. But, even if both materials exhibit similar rate determining step, the larger Tafel slope seen in Ru-MO@Ti indicates a slower OER kinetics when compared to RuO₂@Ti. We have clarified this issue in the main manuscript.

“Likewise, the Tafel slope of Ru-MO@Ti was equal to 234 mV dec⁻¹, such a large value is still associated to the water adsorption being the *rds*, similar to RuO₂@Ti, and likely caused by the low catalyst loading.”

Reviewer #2

In this paper, the authors investigated the synthesis of Sb-Sn-Mo-W mixed-oxide stabilized RuO₂ and studied the OER performance in acidic environments. The mixed oxides can improve the stability of RuO₂ in harsh reaction environments. However, the enhanced stability along with significantly decreased the catalytic activity and increased overpotentials of RuO₂, as shown in Figure 7. Generally, the extended stability may always occur when the catalytic activity is low. Meanwhile, in the paper, the authors did not investigate why these kinds of metal oxides were chosen and how they can promote the stability of RuO₂. As such, I cannot recommend it can be published by Communications Engineering.

Answer: OER electrocatalysts dissolution in PEM electrolysis is a big challenge where a variety of approaches have been reported, as stated in the introduction section, and up to now, there are no clear winners. Commercial OER catalysts in PEM water electrolyzers are still

heavily based on Ru-Ir mixtures, and even those need to be used with high loadings to face the poor stability under harsh conditions.

Our work aims at investigating the feasibility of producing acid-stable oxides capable of hosting OER active metals without hindering their catalytic activity. We selected a quaternary mixed oxide because it is clear that single-oxides or a binary-mixture have not been successful. The importance of higher degree-mixtures is the possibility to form a composite system with unique properties, but how true is this statement? Here we have addressed this, and other issues related to the production of multi metal oxides hosting active OER catalysts, our work has identified the following:

- quaternary mixed oxides can be produced using an already scalable synthesis technique.
- the mixed oxide comprises a well-interconnected network of nanostructured WO_3 , Sb_2O_3 , SnO_2 , and MoO_3 .
- Mo is leached out under acidic and anodic conditions, increasing the porosity of the mixed oxide.
- Ru can be anchored and stabilized in the mixed oxide, improving its resistance against corrosion under anodic potentials and acidic environments.
- Ru is stabilised due to impeded formation of both O vacancies and higher valence ruthenium (Ru^{+8}).
- the degradation of Ru follows a second order reaction bringing a new approach to study corrosion of OER active metals.
- Ru-MO reached 10 mA cm^{-2} at 1.86 V with solely 2.6 % w/w of Ru into the matrix.
- unprotected RuO_2 losses up to 90% of its initial activity when subjected to similar stress tests than that of protected Ru-MO.
- titanium fiber felt (commercial anode gas diffusion layer) is also protected against corrosion when the mixed oxide is used.

With all the findings stated above, our manuscript contributes to the existing literature the following:

- Information about opportunities and challenges when producing acid-stable multi-metallic mixed oxides.
- Proof of concept that mixed oxides can host OER active metals.

- Our approach opens the possibility to stabilize other inexpensive OER active metals, such as Fe, Ni, or Co, inside of the quaternary mixed-oxide matrix.
- Our work shows that multi-metal oxides have the potential to extend the lifetime of the active metal and Ti support when used for water electrolysis in acidic media.
- Conductivity is perhaps the main challenge when using acid-stable mixed oxides.
- A first insight into stabilization mechanisms of Ru into the mixed oxide.
- Highlights the need for studies to understand the complex interactions between all the constituents.

All these findings can aid to go one step forward towards a truly large-scale production of highly stable and active catalysts for water splitting in acidic media, which we believe is of high relevance for the readers of *Communications Engineering*.

Reviewer #3

In this manuscript, the authors reported the quaternary Sn-Sb-Mo-W mixed oxide as protective scaffold for oxygen evolution active metals in acidic media. The findings from this work are interesting and meaningful. I suggest accepting this work after major revisions.

Comment #1. A reduction of 20% in OER activity after 2000 CVs is not satisfied compared to the reported values.

Answer: The observed reduction of 20% is only achieved when the electrocatalyst is stress under cycling voltammetry due to the constant change in oxidation state of the metals comprising the electrocatalysts, and thus a larger degradation is typically observed when compared to chronopotentiometric test. We therefore performed a chronopotentiometric test at 2.5 mA cm^{-2} in a set of new Ru-MO@Ti and RuO₂@Ti samples, the results are shown below in **Figure R1**. We aim here at comparing once again the stability of Ru-MO@Ti and RuO₂@Ti. From **Figure R1**, we observed that after 10 h of continuous testing, Ru-MO@Ti exhibit minimum degradation (only an increase of 10 mV was observed). On the other hand, the reference RuO₂@Ti exhibited a notably corrosion with an increase in potential to 500 mV. From these results we see that the stability is significantly improved when embedding the Ru into the MO scaffold. This conclusion is similar to the one reached after the 2000 CV-test depicted in **Figure 7c** in the main manuscript.

We have added **Figure R1** to supporting information, and a relevant discussion to the main manuscript as seen below:

“Following the stability test, a final polarization curve was recorded in the potential window of 1.2-2V vs RHE. In **Figure 7d**, the RuO₂@Ti exhibits an additional passivation between 1.7 and 1.9V vs RHE, contrary to the Ru-MO@Ti coating where no additional features are observed. These results indicate that the MO coating protects both the Ru and the Ti substrate.

Similar conclusions are obtained when performing a chronopotentiometry study (2.5 mA cm⁻² for 10 h, see **Figure S10**) instead of the 2000-CV test. After 10 h of operation, Ru-MO@Ti experienced only a minor increase in potential of 10 mV, while RuO₂@Ti exhibited a significant degradation with an increase of 500 mV after 10 h. These results highlight the importance of the Sn-Sb-Mo-W mixed-oxide as an effective and protective environment for Ru, reducing its corrosion and extended the operational stability during OER in acidic media.”

Figure R1. Chronopotentiometry study of Ru-MO@Ti and RuO₂@Ti at 2.5 mA cm⁻² at pH=0.

Comment #2: The roles of WO₃, Sb₂O₃, SnO₂, and MoO₃ should be further highlighted.

Answer: Thanks for the suggestion, we have now highlighted role of WO₃, Sb₂O₃, SnO₂ and MoO₃. We have the following information:

Conclusions: “The acid-stable oxide was produced as a coating using solution precursor plasma spraying creating a well-interconnected network of nanostructured WO₃, Sb₂O₃, SnO₂,

and MoO₃. The role of these oxides is to create a stable scaffold where Ru can be anchored, improving its resistance against corrosion under anodic potentials and acidic environments, the latter is possible due to impeded formation of O_{vac} as well as higher valence ruthenium (Ru⁺⁸). The Sb-Sn-Mo-W mixed-oxide coating deposited on FTO glass showed indeed excellent stability in a wide range of acidic conditions (from pH=2 to 0), finding that only MoO_x was partially dissolved leading to a porous structure.”

Comment #3: What about the solubility of WO₃, Sb₂O₃, SnO₂, and MoO₃ during the acid OER process?

Answer: According to Pourbaix diagrams both WO₃ and SnO₂ are stable under OER conditions. In particular, WO₃ is stable at pH < 4 and potentials above 0.5 V vs SHE. (Youngsoon Kim *et al* 2005 *J. Electrochem. Soc.* **152** C89, M. Anik and K. Osseo-Asare 2002 *J. Electrochem. Soc.* **149** B224). SnO₂ is stable at low pH and anodic potentials [Geiger, S., *et al. Sci Rep* **7**, 4595 (2017)]. While Sb₂O₃ is stable under acidic conditions but at anodic potentials (~ 0.7 V vs SHE) a transformation to insoluble Sb₂O₅ can occur. (Arthur L. Pitman *et al* 1957 *J. Electrochem. Soc.* **104**594). However, this transition was not detected after the electrochemical stress indicating that we have an interconnected network of metal oxides that limits any significant chemical changes. Therefore, WO₃, Sb₂O₃, SnO₂ are all stable and insoluble in acidic media. However, MoO₃ forms soluble derivatives under anodic conditions at pH < 2.

All this is in agreement with the observed results. In **Figure 4** we report the corrosion of elements present in the MO coating. As discussed, W, Sn, and Sb exhibited high resistance against corrosion along all pH conditions and potential range tested in this work. While Mo is the only element that significantly dissolved after the OER test.

We have clarified the solubility of the metals involved in the scaffold.

“After the 500-CV test, the Sb/Sn ratio remained nearly constant along the studied pH range, indicating limited corrosion as expected from their theoretical Pourbaix diagrams.^{31,32} However, as the pH of the electrolyte was decreased, the W/Mo ratio increased up to ~5.5 in both EDX and XPS analysis. Since WO₃ is stable under the selected conditions,³³ the increased in W/Mo ratio indicates that Mo was dissolved during the electrochemical stress test as expected from its Pourbaix diagram.³⁴”

As an additional note, we already had the following paragraph in the discussion section:

“The ratio W/Sb (**Figure S4**) was found close to 3.5 after every electrochemical stress test, confirming that the content of W, Sb, and Sn remained nearly constant under the studied pH conditions.”

Comment #4: The reported current density is low and how to fulfill the requirements of large-scale application of PEM water electrolysis?

Answer: We are aware of the low current density seen in our Ru-MO catalyst, and we attribute the low current density to two main factors: (i) Low Ru content, and (ii) poor electrical conductivity of the MO matrix. These factors can be addressed by increasing the Ru loading to improve both conductivity and availability of active sites. But this approach is still limited by the usage of expensive and scarce Ru. So, alternatively we need to improve the conductivity of the MO matrix by other means such as doping and phase engineering, and then stabilize other cheaper OER active metals, ideally Fe, Ni, or Co. However, this should be thoroughly investigated in subsequent studies. In our manuscript, we report the stabilization of Ru as OER active metal, and this approach could be expanded to host and stabilize other OER active metals.

We already mentioned these limitations in the discussion section (see the text below), but we have also highlighted these in the conclusions, and added the approach of introducing other inexpensive OER active metals:

Paragraph already in the manuscript: “From these results two key characteristics are worth noticing: (i) the low current density, and (ii) the large OER onset potential ... The small amount of Ru incorporated into Ru-MO reduced nearly eight times the sheet resistance when compared to pristine MO. It has been reported that the addition of at least 20% w/w of noble metals to unary metal oxides is required to significantly improve both conductivity and OER activity.^{35,36} This contrasts with the 2.8% w/w Ru content in the precursor solution used during the production of our electrodes. Nonetheless, at this stage we aim at studying the incorporation and stabilization of Ru into a matrix of stable oxides to produce a durable electrocatalyst for acidic media electrolysis, while the increase in conductivity without affecting electrochemical stability is a work in progress.”

Conclusions: “Besides, the catalyst supported on titanium fiber felt led to an ECSA of 131.4 cm² associated to a mass activity of 4440 A g⁻¹_{Ru} at 1.8 V (vs RHE), a highly competitive value compared to other Ru-based state-of-the-art electrocatalysts. Our approach also opens the possibility to stabilize other inexpensive OER active metals, such as Fe, Ni, or Co, inside the MO matrix. Finally, we have shown that multi-metal oxides have the potential to extend the lifetime of RuO₂ and Ti-based gas diffusion layers during OER in acidic media, which is one of the main limitations for industrial-scale renewable hydrogen production.”

Comment #5: The following papers (Nature Nanotechnology, 2021 16, 1371–1377; Molecules, 2021, 26, 18, 5476; Energy & Environmental Materials, 2022, e12441) are recommended to be cited for improving the manuscript.

Answer: We have added the suggested references in the introduction section.

REVIEWERS' COMMENTS:

Reviewer #1 (Remarks to the Author):

The authors responded appropriately to the comment given by the reviewer. However, too low current density of a PEMWE in this study should be improved in future work. Please shortly mention strategy to increase real cell performance.

Reviewer #2 (Remarks to the Author):

After carefully reading the revised paper, the quality is improved a lot. It can be accepted. For my comment regard to the role of different oxides, maybe it is difficult to understand fully at the current stage. But it is still worthy to further investigate.

Reviewer #3 (Remarks to the Author):

It can be accepted now.

UMEÅ UNIVERSITY

Rebuttal Letter

Reviewer #1: The authors responded appropriately to the comment given by the reviewer. However, too low current density of a PEMWE in this study should be improved in future work. Please shortly mention strategy to increase real cell performance.

Answer to Reviewer #1. We completely agree that the low current density should be addressed in future works, and from this study the possible strategies to follow are: (i) Increase the amount of Ru into the quaternary oxide. In our study we only use 2.8 % w/w of Ru, while other studies have reported that 20% w/w of Ru is required to achieve a decent conductivity. (ii) Increase the surface area. This can be done by adding additional sacrificial agents (Mo, Ca, or Al) to increase the porosity of the coating. (iii) Increasing the metallicity of the MO by promoting the formation of small band gap oxides such as antimonates.

Strategy (i) and (ii) were already in the manuscript in the section “Incorporation of Ru to MO”. We have added (iii) into the same section. The changes are shown below and highlighted in the manuscript.

“From these results two key characteristics are worth noticing: (i) the low current density, and (ii) the large OER onset potential. These can be caused by the limited conductivity of the FTO glass, the low-Ru content in the electrode, and the substantially low conductivity of the Ru-MO coating. The sheet resistance of MO and Ru-MO@24h are $12.2 \text{ M}\Omega \text{ sq}^{-1}$ and $1.5 \text{ M}\Omega \text{ sq}^{-1}$, respectively. The small amount of Ru incorporated into Ru-MO reduced nearly eight times the sheet resistance when compared to pristine MO. It has been reported that the addition of at least 20% w/w of noble metals to unary metal oxides is required to considerably improve both conductivity and OER activity.^{41,42} This contrasts with the 2.8% w/w Ru content in the precursor solution used during the production of our electrodes. Therefore, the conductivity could be improved by increasing the amount of Ru, or by promoting the formation of conductive or small-band gap metal oxides. Nonetheless, at this stage we aim at studying the incorporation and stabilization of Ru into a matrix of stable oxides to produce a durable electrocatalyst for acidic media electrolysis, while the increase in conductivity without affecting electrochemical stability is a work in progress.”

Reviewer #2: After carefully reading the revised paper, the quality is improved a lot. It can be accepted. For my comment regard to the role of different oxides, maybe it is difficult to understand fully at the current stage. But it is still worthy to further investigate.

Answer to Reviewer #2. We agree that identifying the role that each metal has in the MO matrix is of great importance to be able to tackle the weakness of the produced MO, such as conductivity, and low OER activity. This is something that we aim at addressing in the near future. No changes were made in the manuscript.

UMEÅ UNIVERSITY

Reviewer #3: It can be accepted now.

Answer to Reviewer #3: No changes were required.